# Investigating White Matter Abnormalities Associated with Schizophrenia Using Deep Learning Model and Voxel-Based Morphometry

**DOI:** 10.3390/brainsci13020267

**Published:** 2023-02-04

**Authors:** Tripti Goel, Sirigineedi A. Varaprasad, M. Tanveer, Raveendra Pilli

**Affiliations:** 1Biomedical Imaging Lab, Department of Electronics and Communication Engineering, National Institute of Technology Silchar, Silchar 788010, Assam, India; 2Department of Mathematics, Indian Institute of Technology Indore, Simrol 453552, Madhya Pradesh, India

**Keywords:** magnetic resonance imaging, random vector functional link, Schizophrenia, voxel-based morphometry

## Abstract

Schizophrenia (SCZ) is a devastating mental condition with significant negative consequences for patients, making correct and prompt diagnosis crucial. The purpose of this study is to use structural magnetic resonance image (MRI) to better classify individuals with SCZ from control normals (CN) and to locate a region of the brain that represents abnormalities associated with SCZ. Deep learning (DL), which is based on the nervous system, could be a very useful tool for doctors to accurately predict, diagnose, and treat SCZ. Gray Matter (GM), Cerebrospinal Fluid (CSF), and White Matter (WM) brain regions are extracted from 99 MRI images obtained from the open-source OpenNeuro database to demonstrate SCZ’s regional relationship. In this paper, we use a pretrained ResNet-50 deep network to extract features from MRI images and an ensemble deep random vector functional link (edRVFL) network to classify those features. By examining the results obtained, the edRVFL deep model provides the highest classification accuracy of 96.5% with WM and is identified as the best-performing algorithm compared to the traditional algorithms. Furthermore, we examined the GM, WM, and CSF tissue volumes in CN subjects and SCZ patients using voxel-based morphometry (VBM), and the results show 1363 significant voxels, 6.90 T-value, and 6.21 Z-value in the WM region of SCZ patients. In SCZ patients, WM is most closely linked to structural alterations, as evidenced by VBM analysis and the DL model.

## 1. Introduction

Schizophrenia (SCZ) is a severe mental condition marked by the abnormal perception of reality, hallucinations, delusions, and profoundly disordered thoughts and behavior that causes significant impairment in daily life. SCZ patients need to be monitored by a doctor for the rest of their lives. For better results, diagnosis and therapy should begin as soon as possible, before the condition reaches its severe stage [1,2]. SCZ patients are diagnosed by collecting a thorough patient history, assessing their mental state through a comprehensive physical examination, and conducting any necessary laboratory tests. Aside from clinical rating scales, researchers are working hard to identify imaging biomarkers that can help with the diagnosis, treatment, and prognosis of illnesses and their subtypes.

Recently, neuroimaging studies have uncovered anomalies in SCZ related to structural and functional changes in the brain’s cortical (such as the frontal region), subcortical (such as the hippocampus, thalamus), and network connectivity areas [3,4,5]. Particularly, there is growing enthusiasm for utilizing structural neuroimaging findings to better diagnose SC. Gray Matter (GM), White Matter (WM), and Cerebrospinal Fluid (CSF) are the three primary regions of the human brain. Over half the human brain is composed of WM. WM is made up of bunches of myelinated axons that connect neurons to various brain regions. GM comprises neuronal cell bodies as well as dendrites that are localized in the cortex’s outer layers. CSF in the brain and spinal cord is a clear, colorless fluid. CSF has many purposes, including the prevention of cerebral ischemia and the protection of the brain from harm. Many advancements in MRI acquisition and image processing have occurred, including functional MRI (fMRI) and diffusion tensor imaging (DTI), all of which have empowered us to fully exploit the information contained in the human brain. GM abnormalities in SCZ have been investigated and detected using structural magnetic resonance imaging (sMRI) [6]. With the development of DTI, we can also investigate WM abnormalities in SCZ [7]. Numerous neuropathological abnormalities have been observed in SCZ patients, and the latest research reveals that the progression of the disease may be significantly influenced by the neurological substrates of connection [8]. Walterfang et al. [9] reviewed WM pathology as a potential substrate of poor connection in SCZ and investigated the relevant evidence. However, while conventional structural techniques such as DTI can effectively probe the inner workings of WM architecture, they cannot reveal the neuronal activity and important functions within WM. The fMRI, which is based on blood-oxygen-level-dependent (BOLD) data, has shown to be an effective tool for studying the functional organization of the human brain in the fields of cognitive neuroscience and clinical neuropsychiatry [10].

Recent studies have found that WM functional activity relates to demands in a range of activities, including perceptual, linguistic, and motor tasks [11]. Schlosser et al. [12] employed the combination of fMRI and DTI to demonstrate the altered structure–function relationships in SCZ. According to DTI analysis, there were decreases in fractional anisotropy (FA) in the right frontal lobe and the right medial temporal lobe, which are regions thought to include inferior cingulum bundle fibers. Second, among patients with a primary task-related effect, fMRI revealed significant hypoactivation in the prefrontal, superior parietal, and occipital areas. Along with the reduction in frontal FA, the patient’s fMRI activation in the prefrontal and occipital cortical regions was reduced. Jiang et al. [13] evaluated the functional connectivity of WM with a large cohort of 97 SCZ and 126 cognitive normal subjects. In this research, the authors performed a cluster analysis to determine voxel-based WM functional connectivity and classified the ten largest WM networks into three distinct layers: superficial, intermediate, and deep. Individuals with SCZ showed reduced low-frequency oscillation amplitudes and increased functional connectivity in the motor networks involved in superficial perception, according to an analysis of spontaneous oscillation and its functional connectivity in the WM network. Singh et al. [14] analyzed the correlation between hemodynamic and morphometric measurements to ascertain whether the lack of motor function in SCZ patients is connected to anatomical anomalies in particular brain regions. In other tests, fMRI scans were conducted on subjects and controls who had SCZ while they were tapping their right index and middle fingers. During a straightforward finger-tapping test, patients with SCZ displayed decreased activity in the ipsilateral and contralateral motor areas compared to controls.

With the advancements of machine learning (ML) and the subsequent proliferation of computer-assisted diagnosis, MRI has found widespread application in recent years for SCZ diagnosis [15]. The application of ML classifiers for SCZ diagnosis, however, requires first extracting features from MRI data [16]. To enable a computer to learn from data, ML creates algorithms that are related to artificial intelligence [17]. By using prior knowledge, analysis, and self-training, ML enables computers to deal with novel situations. These techniques do not perform well with raw data. Thus, the features must be manually extracted. Handcrafted feature-based ML techniques impose limitations on the usage of computer-aided diagnostics (CAD) in real applications because the chosen features might not be reliable. Deep learning (DL) algorithms with various designs have gained popularity because of their capacity, accuracy, and efficiency in resolving a wide range of problems [18]. DL algorithms’ clinical relevance has improved for specific diagnostic applications, such as the detection of pulmonary nodules using chest computed tomography (CT) scans [19], diagnosis of Alzheimer’s Disease using MRI images [20], and the retinal fundus images for the diagnosis of diabetic retinopathy [21], due to their enhanced performance in visual image identification [22]. DL is a type of ML that “learns” by analyzing large amounts of labeled data and “recognizes” key properties [23] without requiring the user to define specific qualities. Various studies have been incorporated using different DL-based approaches using neuroimaging data [24,25]. The benefits of DL algorithms include their ability to handle huge, complicated datasets, their use of spatial correlation, and their numerous convolutional layers. Algorithms for DL can have automatic feature extraction that is crucial for classification. The main issue with traditional DL algorithms is that their classification layer uses back-propagation (BP) algorithms. BP algorithms have several drawbacks, including being time-consuming, depending on the learning rate for convergence, becoming stuck at the local minimum, and other issues.

Single-layer feed-forward neural (SLFN) networks have undergone extensive investigation over the past 20 years and have been incorporated into a range of applications, including classification tasks and regression tasks, due to their capability to mimic any function. Schmidt et al. [26] presented a feed-forward neural model with random weights based on the randomization technique. In addition to directly connecting the input and output layers, Pao et al. [27] implemented the random vector functional link (RVFL) network in which only the output weights must be analytically calculated, and all other parameters are randomly generated from a stable domain. The RVFL’s universal approximator capability was proved by Igelnik and Pao [28]. The number of enhancement nodes and activation functions that will be utilized to train the standard RVFL must be known in advance, and it must overcome the limitations of BP algorithms. Manually determining the optimal concealed node range and the ideal activation function is challenging. Deep and ensemble-deep architectures [29] have been added to the shallow RVFL model to enhance its generalization performance. In the deep RVFL (dRVFL) model, the hidden layer’s parameters are generated at random and maintained constant throughout training, whereas only the parameters of the output layer, which contains multiple stacked layers, must be calculated analytically [30]. Representation learning is improved by comparing the dRVFL model with the shallow RVFL model. When the amount of training data, hidden layer count, and feature dimension are high, the dRVFL model experiences memory problems. Therefore, an implicitly ensemble methodology-based ensemble deep RVFL network (edRVFL) model has been developed to overcome these concerns.

In this study, a pretrained DL network, ResNet-50 [31], is used to extract efficient high-level features from 2D MRI slices. The RVFL network and its derivatives, a type of SLFN, has been used to categorize these characteristics. During training, RVFL’s weights and biases for its enhancement layers are arbitrarily fixed after being selected within a suitable range at random. Voxel-based morphometry (VBM) is a neuroimaging approach that analyses voxel-wise 3D brain scans to discover changes in regional GM, WM, and CSF volumes between two groups of subjects, such as SCZ patients and control normal (CN) subjects. Images from different subjects are normalized and registered to create a brain template to perform VBM [32]. The following are the key contributions of the current study.

Using Statistical Parametric Mapping (SPM), version 12 i.e., SPM12 software, we incorporated preprocessing of the data to increase the model’s learning efficiency and accuracy.A Generative Adversarial Network (GAN) is employed to increase the size of the dataset.ResNet-50 is used to extract robust features from high-dimensional images.The edRVFL classifier is employed to classify the extracted features, and results are acquired using majority voting to obtain efficient outcomes.VBM analysis is performed to evaluate the structural changes related to GM, WM, and CSF volumes.Finally, structural abnormalities related to the brain volumes are investigated and concluded by combining the results obtained from the edRVFL model and VBM analysis.

The novelty of this research is correlating DL and VBM analysis of WM alterations in patients with SCZ. MRI scans are segmented into WM, GM, and CSF to examine the specified structural changes. The high-level characteristics of MRI slices are extracted to reduce the complexity of using a pretrained DL network, and the classification is performed using an edRVFL classifier. Furthermore, VBM analysis was carried out to reveal the primary brain regions affected in SCZ patients. Finally, structural abnormalities related to WM are investigated by combining the results obtained from the edRVFL model and VBM analysis.

The remaining part of this paper is organized as follows: The proposed methodology is described in Section 2. We report our findings and analyze the experiments in Section 3. The discussion and remarks are stated in Section 4. The paper is concluded in Section 5.

## 2. Methodology

This section briefly introduces the proposed methodology for SCZ diagnosis using structural MRI scans. The diagnosis paradigm for SCZ is depicted as a block diagram in Figure 1.

### 2.1. Data Preparation

The structural MRI images of SCZ patients and CN subjects were collected from an open source (https://openneuro.org (accessed on 30 December 2022)) with OpenNeuro Accession Number ds000115 on 14 July 2018. These included 99 participants, which consisted of 58 SCZ patients and 41 CN patients. All the participants were between the ages of 12 and 30, of which 39 were females and 60 were males.

### 2.2. Preprocessing

The preprocessing of data is a crucial stage in most of the DL pipelines to generate accurate results. Figure 2 shows the pipeline of the preprocessing of MRI scans.

Neuroimage preprocessing plays a crucial role in disease diagnosis by revealing flaws, outliers, and missing key information. Incorrect preprocessing would leave these flaws in the model, reducing its predictive power. The first step was to extract the GM, WM, and CSF maps from the MRI scans. Preprocessing for MRI includes normalising to the standard Montreal Neurological Imaging (MNI) space via a diffeomorphic registration technique, eliminating non-brain tissue, and modifying the results. The GM, WM, and CSF images were all smoothed using a Gaussian kernel with an FWHM of 8 mm. All the preprocessing steps were performed using the MATLAB 2021a, SPM12 tool. After preprocessing, 2D slice extraction was employed to obtain the 2D MRI images. For better visualization, we extracted the middle 10 slices out of 256 slices from both SCZ and CN subjects of WM, GM, and CSF.

### 2.3. GAN Architecture

Due to the smaller sample sizes of the gathered dataset, it cannot be used to train a DL network effectively. To address this issue, the proposed model employed GAN to expand the available datasets [33].

The generator and discriminator are the two DL models that makeup the GAN, and they both work together to automatically find and learn patterns from the input data. One competes with the other to find and record unique features in the dataset. New instances can be generated using GANs that are convincing enough to be considered part of the original dataset. The generalized GAN network structure is depicted in Figure 3, and the loss function of GAN is given in Equation (Equation 1).
(1)$$min_{G}max_{D}V\left(D,G\right)=E_{z∼pdata(z)}\left[logD\left(z\right)\right]+E_{x∼p_{x}\left(x\right)}\left[log\left(1-D\left(G\left(x\right)\right)\right)\right]$$

In Equation (Equation 1), *G* and *D* represent the corresponding networks for the generator and discriminator, pdata(z) denotes the distribution of real data, p(x) represents the distribution of the generator, *z* is the sample from pdata(z), *x* is the sample of P(x), D(z) is the discriminator network, and G(x) is the generator network.

Original MRI slices and GAN-generated slices from each region: WM, GM, and CSF, are fed to ResNet-50 [23], a pretrained DL network for the feature extraction. ResNet-50 consists of 50 layers, being the most widely used model for high-level feature extraction from an image. The skip connection is ResNet’s key technological advancement. The ResNet-50 architecture for extracting features from an MRI picture is displayed in Figure 4.

### 2.4. Ensemble Deep Random Vector Functional Link Network (edRVFL)

RVFL is a non-iterative SLFN. The main concept of the RVFL network [28] is to initialize the random parameter for the enhancement layer (hidden layer), and random parameter values are fixed during the training phase. For MRI images, which are slightly more complex in nature, BP-based classifiers may require much more training time. The training phase of RVFL is faster than other BP classifiers since it has a single hidden layer and non-iterative operation. The output layer of RVFL is fed with both original input features denoted as *X* and output features of the enhancement layer represented as *H*, which can be expressed as D=[HX]. If *x* is the total input features and *k* is the number of neurons, then there are (x+k) total features fed to the output layer. The weights *W* and bias *B* are fixed at the enhancement layer, and the output weights β will be calculated. As a result, the optimization function may be stated mathematically as:(2)$$\underset{\beta }{m}{in∥D\beta -O∥}^{2}+\eta {∥\beta ∥}^{2},$$
where η is the regularization parameter, and *O* is the output target.

Equation (Equation 2) is solved by using η≠0 (ridge regression) or η=0 (Moore–Penrose pseudoinverse). If η=0, the solution is **β=D+O**; however, when ridge regression is used, the solution is
(3)$$\beta ={\left(D^{T}D+\eta I\right)}^{-1}D^{T}O.$$

The deep RVFL network contains one input layer, various enhancement layers stacked in parallel, and one output layer. Each stack layer receives input data from the preceding layer’s output. The enhancement layer weights and biases are randomly obtained from a fixed domain, and output weights are only analytically computed. In dRVFL, a stack of *L* enhancement layers is employed, and each layer contains *N* number of hidden nodes. First, hidden layer output is expressed as $H^{\left(1\right)}$ = $g(XW^{\left(1\right)})$. For other layers, L>1 output is $H^{\left(L\right)}$ = $g(H^{\left(L-1\right)}W^{\left(L\right)})$; *g*(.) is the radbas activation function. The mathematical expression between the input and output layer is
(4)$$D=[H^{\left(1\right)}H^{\left(2\right)}\ldots \ldots \ldots H^{\left(L-1\right)}H^{\left(L\right)}X].$$

The dRVFL input-to-output layer is comprised of nonlinear features of stacked hidden layers and original input features, and its output is specified as
(5)$$O=D\beta_{d}.$$

The output weight $\beta_{d}$ is solved by using Equation (Equation 3).

The ensemble deep RVFL (edRVFL) architecture is shown in Figure 5. In edRVFL, rather than training *L* neural networks individually, a single deep RVFL network is trained, and the ensemble has *L*-hidden layers with the same number of hidden nodes. The final output weight $\beta_{d}$ of dRVFL is divided into various small $\beta_{d}$. Each $\beta_{d}$ is considered as a separate model and is determined independently. The ultimate result is acquired by majority voting. Each higher-level model receives a concatenation of the previous model’s non-linearly modified features and original input data features. The edRVFL first enhancement layer output is expressed as
(6)$$H^{\left(1\right)}=g\left(XW^{\left(1\right)}\right).$$

The output of the other enhancement layers L>1 is expressed as
(7)$$H^{\left(L\right)}=g\left(\left[H^{\left(L-1\right)}X\right]W^{\left(L\right)}\right).$$

The output weights $\beta_{ed}^{\left(1\right)}$, $\beta_{ed}^{\left(2\right)}$, …, $\beta_{ed}^{\left(L\right)}$ are then calculated independently using Equation (Equation 3).

### 2.5. Voxel-Based Morphometry (VBM)

VBM [32] is a whole-brain impartial, objective method used to evaluate changes in the brain in real time using MRI data. VBM looks for differences in the way brain tissues are made in different parts of the brain. The MRI scans must first be segmented to reveal the GM, WM, and CSF regions. The GM, WM, and CSF volumes are then spatially normalised to the MNI space. After that, statistical analysis is performed to determine whether or not there are statistically significant differences between SCZ patients and CN. The native space volumes of GM, WM, and CSF images are derived from total intracranial volume (TIV), which serves as a reference in this investigation. After establishing a p<0.05 threshold and conducting family-wise error (FWE) correction, a two-sample *t*-test was executed. Finally, we used the xjview MATLAB software to capture the voxel brain region (shown in pseudocolour), substantial changes, activation volume (cluster), and activation intensity. We employed a voxel extent threshold of 0.02 (statistically analyzed with two sample *t*-test and expressed as T value; the T value is proportional to the intensity). The processing framework for VBM analysis is depicted in Figure 6.

## 3. Performance Evaluation of the edRVFL Model and VBM Analysis

This section discusses the experimental results and their comparison with leading-edge classifiers. Utilizing the DL model and VBM analysis, this portion also looked into the relationship between WM and SCZ.

### 3.1. Implementation Detail

The proposed work was carried out in MATLAB 2021a and executed on a Windows 10 computer with an Intel (R) Xenon (R) W-2133 processor running at 3.60 GHz and 64 GB RAM. NVIDIA Quadro P2200 was the GPU in use. Using a publicly available dataset, the suggested model was evaluated for CN subjects and SCZ patients. The dataset was split into training and testing sets in the ratio of 70:30. The whole data preprocessing steps were conducted using the SPM12 toolbox. After preprocessing and slice extraction, the data obtained were not sufficient to be used effectively for training the DL network. This issue can be addressed by incorporating a GAN for image generation. Using the GAN network, data size was increased from 410 CN and 580 SCZ for WM, GM, and CSF to 1000 CN and 1000 SCZ slices for WM, GM, and CSF. For RVFL networks, the number of hidden units were tuned from the range $\left\{256,512,1024\right\}$ for each GM, WM, and CSF dataset. The regularization parameter η of the RVFL models was set to $\frac{1}{C}$, and *C* was selected from the range $2^{z}$, where $z=\left\{-6,-4,\ldots ,12\right\}$. The enhancement layers were considered to be five. The “radbas” activation function was used in all experiments, and the two-stage tuning method was used to obtain optimal hyperparameter configuration [34].

### 3.2. Performance Metrics

In this segment, we will discuss the performance metrics used to evaluate the model’s capability to diagnose SCZ patients. Accuracy (Acc), sensitivity (Sens), specificity (Spec), precision (Prec), recall (Rec), F-score, G-mean, confusion matrix (CM), and receiver operating characteristic (ROC) were determined for the given model in order to evaluate its capabilities. Specificity assesses the classifier’s ability to identify disease-free individuals reliably. Sensitivity measures the classifier’s ability to diagnose the person with the disease correctly. Precision measures the ratio of true diseased out of predicted diseased subjects. Accuracy can be elaborate as the true revealer of the full process. G-mean examines the balance between classification’s performances on the multiclass that is provided. CM shows the prediction analysis and helps to calculate the performance metrics necessary to deduce the conclusion. The ROC curves are sensitivity vs. specificity 2D plots. The curves near the left corner of the top will achieve good results.

### 3.3. Computational Complexity and Model Parameter Sensitivity Analysis

The computational complexity (CC) of the current approach combines the CC of ResNet-50 and edRVFL. Training a ResNet-50 has a CC of $O(MKn^{2}FA)$, where *M* and *K* are the dimensions of a 2D image, n×n is the kernel filter size, *F* is the number of filters, and *A* represents the number of activations. In RVFL networks, matrix inversion is used to determine the output weights. The input training data dimensions define the computation needed for calculating the pseudoinverse. The computations needed to calculate the output for RVFL of N×N matrix size is $O({\left(N+k\right)}^{3})$ time, where *k* represents the number of hidden nodes. In deep RVFL, the CC is $O({\left(kL+m\right)}^{3})$ time, where *L* is hidden layers, and *m* is the dimension of the input data. In edRVFL, we decompose the overall output weight $\beta_{d}$ of deep RVFL into various small $\beta_{ed}$. Each small $\beta_{ed}$ is assessed individually, and the final outcome is decided by majority vote or model averaging. Every $\beta_{ed}$ requires the matrix inversion of size ((k+m)(k+m)) for the first layer and for higher layers ((2k+m)(2k+m)).

### 3.4. Comparison of Different Regions of the Brain

Table 1 depicts the performance analysis of WM with different regions of the brain area using ResNet-50 as the feature extractor and edRVFL as the classifier. Figure 7 and Figure 8 show the comparison using the ROC and CM for WM, GM, and CSF. The proposed model achieved 96.50%, 88.17%, and 89.17% classification accuracy with WM, GM, and CSF datasets, respectively. Based on Acc, Sens, Spec, Prec, Rec, F-score, and G-mean, the WM was found to be highly affected.

### 3.5. Comparison with Different State-of-the-Art Classifiers

Table 2 depicts the performance analysis of WM using edRVFL and compares it with state-of-the-art classifiers: K-nearest neighbors (KNN) [35], random forest (RF) [36], decision tree (DT) [37], ensemble bagging (EB) [38], softmax [39], support vector machine (SVM) [40], extreme learning machine (ELM) [41], kernel ridge regression (KRR) [42], RVFL [43], and dRVFL [29]. The edRVFL network based deep model outperformed the different classifiers in terms of Acc, Sens, Spec, Prec, Rec, F-score, and G-mean.

### 3.6. Voxel-Based Morphometry Analysis

As shown in Table 3, the two-sample *t*-test was used with a covariate of TIV to compare SCZ patients to CN. By employing family-wise error (FWE) with p<0.05 in voxel by voxel analysis, three regions: left cerebrum extranuclear, right cerebrum temporal lobe, and left cerebrum claustrum, in the SCZ participants showed more variations in GM volumes than for the CN subjects. WM alterations were found extensively in the right cerebrum insula, right cerebrum temporal lobe, and left cerebrum internal ventricle. We observed no significant brain CSF alterations in SCZ patients over the controls. For WM maps, T-value (6.90) and Z-value (6.21) were more compared with GM maps, which indicates that the WM abnormalities are more in SCZ patients over controls. Figure 9 shows the anatomical changes in WM volume of brain regions.

## 4. Discussion

The goal of the current study was to categorize SCZ based on the kinds of changes that happen to brain tissue. Our most important observation is that SCZ has a stronger relationship with WM than with GM and CSF. Kadry et al. [39] proposed an automatic SCZ detection framework based on MRI scans. The pretrained VGG-16, DL network was used to analyze brain MRIs. The deep features retrieved were optimized using the slime mould algorithm (SMA) and then classified with an SVM binary classifier with 90.33% accuracy. The GM and WM volumes of 42 CN individuals and 41 SCZ patients [44] were analyzed using SVM with 88.4% accuracy. The authors concluded that particular brain neuroimaging patterns related to SCZ might be discovered as a potential biomarker for disease detection. Pinaya et al. [45] utilized a deep belief network (DBN) to extrapolate and interpret characteristics from MRI data from 83 CN subjects and 143 patients with SCZ. The classification accuracy of DBN was 73.6%, which is higher than that of the traditional SVM (68.2%). A DL algorithm [46] was fed with MRI scans that are able to detect SCZ accurately in a random sample of pictures with an AUC of 0.96. The DL algorithm accurately identified SCZ by analyzing structural brain MRI data and isolating relevant structural features. Supriya et al. [47] implemented a 3D CNN framework to detect SCZ over CN subjects. The robust features were extracted using a 3D CNN and classified by an ensemble bagging network with 92.22 % classification accuracy.

In the present study, 99 MRI scans were acquired from the Open Neuro database, out of which 58 were SCZ patients, and 41 were CN subjects. The SPM12 toolbox was used for the segmentation and preprocessing of all MRI images. For the suggested model, we compiled three datasets: GM, WM, and CSF. ResNet-50 extracts features from input 2D slices, and edRVFL performs the classification of the extracted features. Regarding classification accuracy, the model was trained and tested on GM, WM, and CSF and obtained 88.17%, 96.50%, and 89.17% accuracy, respectively. We compared the proposed edRVFL-based deep model to other classifiers such as KNN, RF, DT, EB, softmax, SVM, ELM, KRR, RVFL, and dRVFL on the WM dataset. As can be seen in Table 2, the edRVFL-based deep model outperformed the other classifiers in terms of performance measures.

Voxel-based morphometry (VBM) helps to find differences in the brain’s structure in neurodegenerative diseases such as dementia, Parkinson’s disease (PD), SCZ, and multiple sclerosis (MS). When looking for volumetric differences between GM, WM, and CSF, VBM examines each voxel size of segmented tissue. VBM is not partial to alterations in any one region of the brain; it eliminates the arbitrary distinctions in perception that come from highlighting certain areas of the brain. Qiang Li et al. [48] used VBM and functional connectivity density (FCD) analyses to reveal GM volume reduction and structural and functional abnormalities in the brain. In this study, 79 CN subjects and 55 SCZ patients were used to perform the experiments. The authors identified GM volume decline in temporal, frontal, occipital, and parietal lobes and increased FCD in the cerebellum, decreased FCD in the precuneus, and no GM volume variations in the cerebellum and precuneus. Zhao et al. [49] used diffusion tensor imaging (DTI) and VBM multimodal analysis to identify the common and specific abnormalities in WM volume and fractional anisotropy (FA) between SCZ and bipolar disorder (BD). They discovered that SCZ patients have more significant WM alterations than BD patients. The bilateral corpus callosum had low WM volume and FA in SCZ and BD. Chen Li et al. [50] explored the GM volume alterations in 86 SCZ patients compared with 86 CN subjects using the VBM analysis, and they observed the increased GM within the cerebellum. The limitation of this work is that only default parameter settings are used in VBM processing phases as provided by the development tools. Lee et al. [51] investigated the abnormalities in GM and WM between SCZ patients and BD cases. Some 65 SCZ patients, 65 BD, and 65 CN were enrolled. The authors used VBM analysis for MRI scans, and tract-based special statistics (TBSS) for DTI scans acquired at a single center. They detected GM volume loss in the thalamus and insular and WM abnormalities in the superior longitudinal fasciculus, corpus callosum, external capsule, internal capsule, and posterior thalamic radiation. When comparing SCZ patients and CN participants, we found substantial differences in WM volume in three regions and GM volume in three regions, but no differences in CSF volume as illustrated in Table 3.

### Remarks

For the proposed model, Table 1 depicts the performance analysis of WM with different regions of the brain, using ResNet-50 as the feature extractor and edRVFL as the classifier. According to the results obtained, WM reached 96.50% accuracy compared to CSF’s (89.17%) and GM’s (88.17%). Table 2 depicts the performance evaluation of WM using edRVFL and compares it with state-of-the-art classifiers. In this comparison, edRVFL outperformed with 96.50% accuracy over other state-of-the-art networks such as KNN (94.25%), RF (94.50%), DT (94.25%), EB (94.75%), softmax (94%), SVM (93.50%), ELM (93.33%), KRR (94.33%), RVFL (89.67%), and dRVFL (91.17%). Table 3 depicts voxel-wise variations in WM, GM, and CSF volumes. In this observation, WM shows 1363 significant voxels, a 6.90 T-value, and a 6.21 Z-value in the left cerebral internal ventricle compared to GM and CSF. According to the findings of the above experiments, WM plays a significant role in SCZ patients. As a result, the proposed model may help clinicians diagnose SCZ patients with structural anomalies in the WM.

## 5. Conclusions

This study uses MRI data to train a deep learning (DL) algorithm to create a diagnosis model for schizophrenia (SCZ) patients and control normals (CN). In order to conduct a region-by-region analysis on an MRI scan, the image was first segmented into its component parts: Gray Matter (GM), White Matter (WM), and Cerebrospinal Fluid (CSF). The first step was preprocessing, which involved removing outliers, realigning the image, registering it to the reference template, and extracting the 2D key slices. Due to insufficient data for training a deep learning (DL) model, a generative adversarial network (GAN) was used to produce synthetic pictures. ResNet-50 was used to extract features from the MRI slices, and the features were then classified using an ensemble-based deep random vector functional link (edRVFL) classifier. When using WM volume, the best detection performance was seen for SCZ. A voxel-based morphometry study also showed that SCZ had the most effect on the WM. This leads us to the conclusion that the WM is the primary site of damage in SCZ.

Future work aims to acquire a more extensive dataset from several test sites to implement the suggested architecture into real-world clinical decision making. Additionally, we can investigate functional MRI (fMRI) scans in addition to structural MRI (sMRI) to provide the model with as much data as possible to use when deciding on a course of action.

## Figures and Tables

**Figure 1 brainsci-13-00267-f001:**
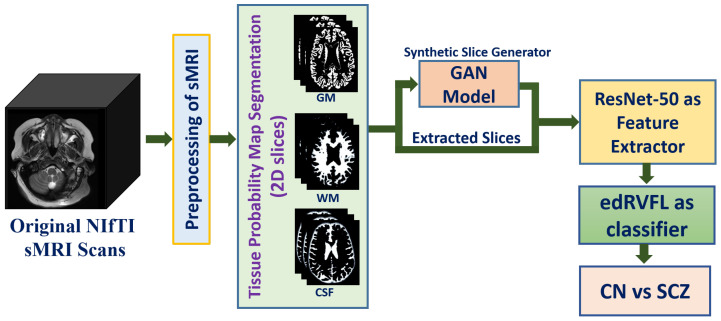
Proposed method for SCZ diagnosis using DL: Neuroimaging Informatics Technology Initiative (NIfTI) images are preprocessed and segmented, and slices are extracted and augmented by a GAN. The robust features from 2D images are extracted and classified by the edRVFL network.

**Figure 2 brainsci-13-00267-f002:**
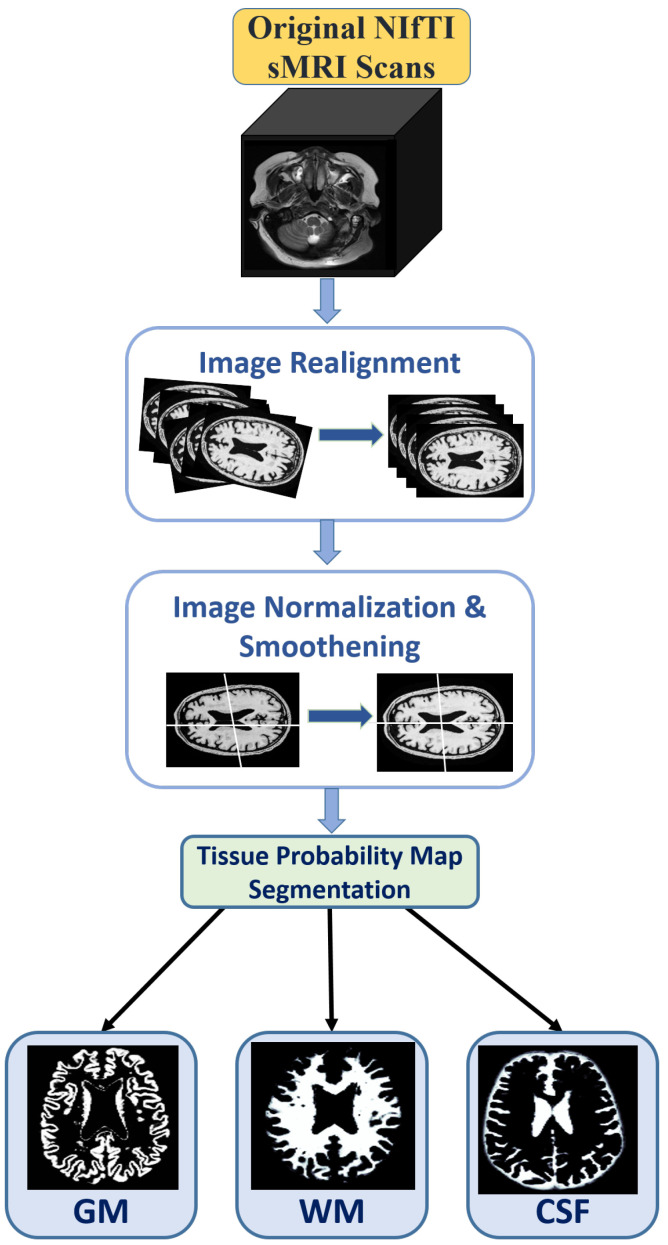
Preprocessing steps involved in dataset preparation are image realignment, normalization, smoothing, and finally segmentation into three tissue maps.

**Figure 3 brainsci-13-00267-f003:**
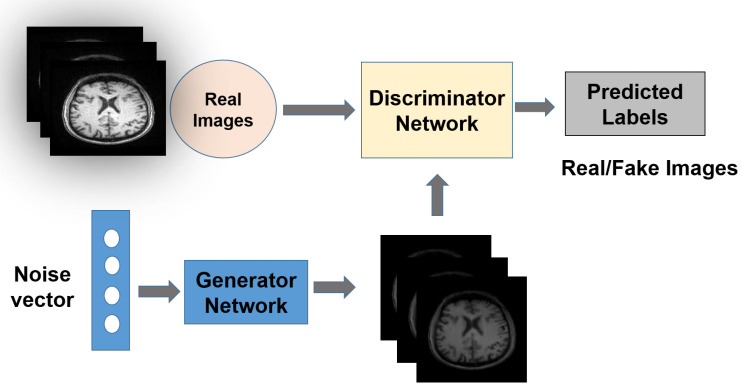
Generative adversarial network architecture: generator and discriminator networks perform adversarial functioning with each other.

**Figure 4 brainsci-13-00267-f004:**
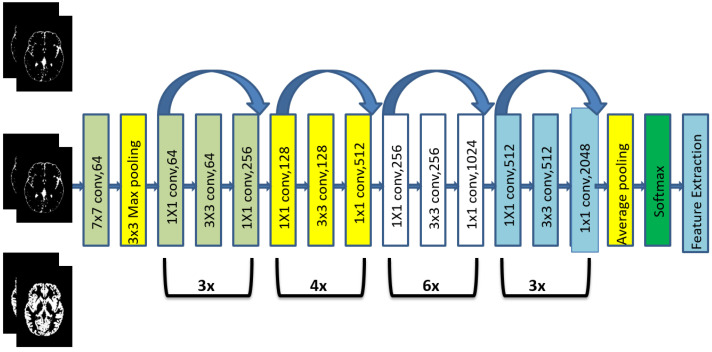
Architecture of ResNet-50.

**Figure 5 brainsci-13-00267-f005:**
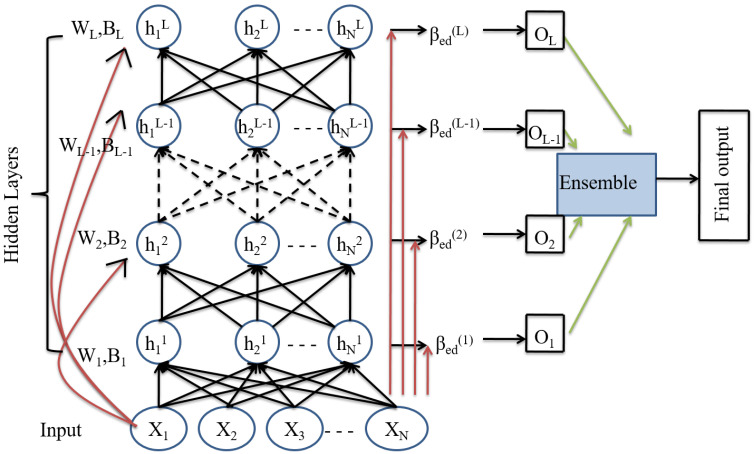
Architecture of ensemble deep RVFL (edRVFL) network.

**Figure 6 brainsci-13-00267-f006:**
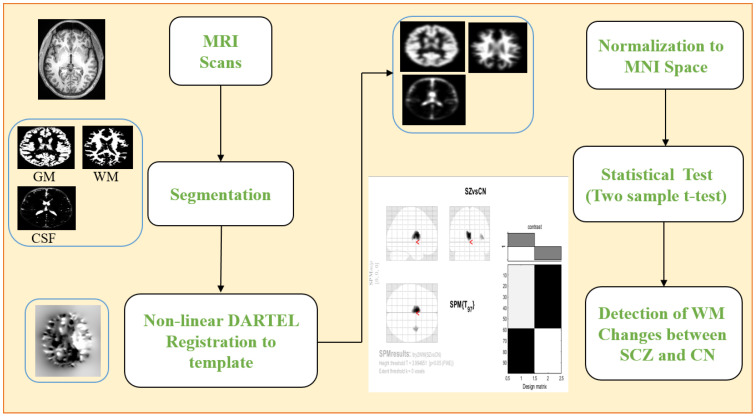
The architecture of the VBM framework using SPM12 software.

**Figure 7 brainsci-13-00267-f007:**
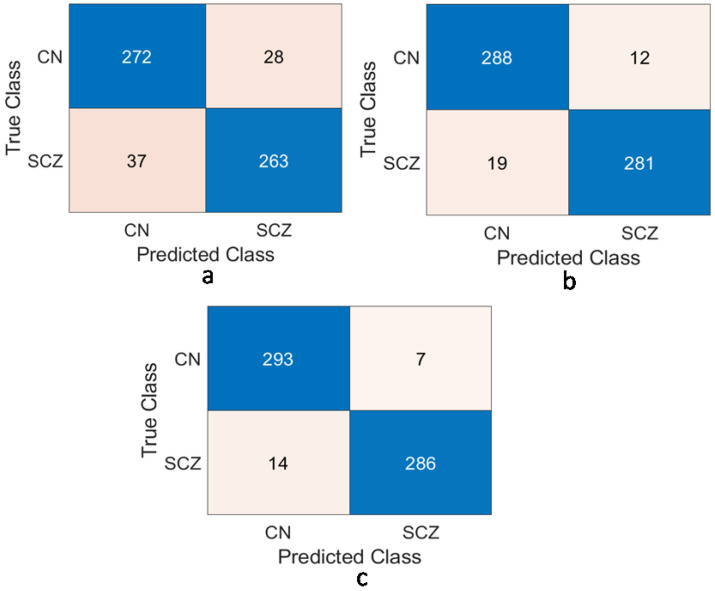
Confusion matrix of: (**a**) CSF-89.17% accuracy; (**b**) GM-88.17% accuracy; (**c**) WM-96.50% accuracy.

**Figure 8 brainsci-13-00267-f008:**
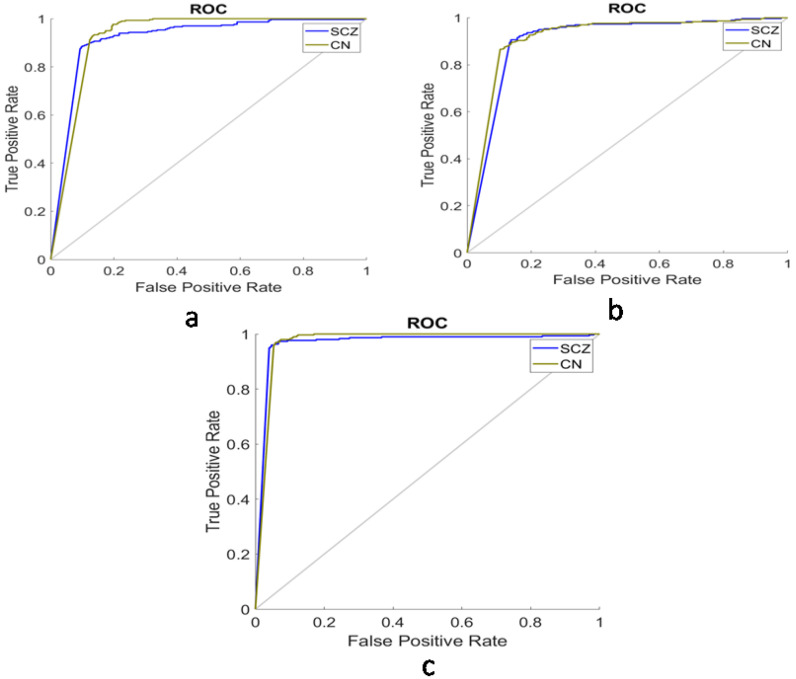
Region of convergence of: (**a**) CSF; (**b**) GM; (**c**) WM.

**Figure 9 brainsci-13-00267-f009:**
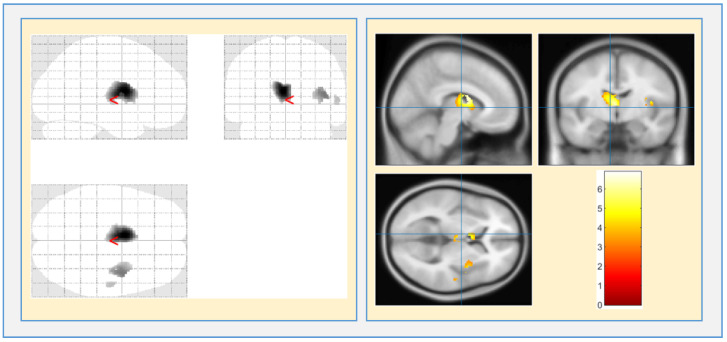
WM volume alterations in the brain regions with p< 0.05 and external threshold K = 0.2.

**Table 1 brainsci-13-00267-t001:** Comparison of WM with other regions of the brain using the edRVFL classifier.

Region of Interest	Acc	Sens	Spec	Prec	Recall	F-Score	G-Mean
Cerebrospinal Fluid	89.17	87.67	90.67	90.38	87.67	89	89.15
Gray Matter	88.17	89.67	86.67	87.06	89.67	88.34	88.15
White Matter	96.50	95.33	97.67	97.61	95.33	96.46	96.49

**Table 2 brainsci-13-00267-t002:** Comparison of WM with various state-of-the-art classifiers.

Classifier	Acc	Sens	Spec	Prec	Rec	F-Score	G-Mean
KNN [35]	94.25	93.50	95	94.92	93.50	94.21	94.25
RF [36]	94.50	93	96	95.88	93	94.42	94.49
DT [37]	94.25	93.50	95	94.92	93.50	94.21	94.25
EB [38]	94.75	93.50	96	95.90	93.50	94.68	94.74
Softmax [39]	94	93.50	94.50	94.44	93.50	93.97	94
SVM [40]	93.50	92.50	94.50	94.39	92.50	93.43	93.49
ELM [41]	93.33	92.67	94	93.92	92.67	93.29	93.33
KRR [42]	94.33	92	96.67	96.50	92	94.20	94.30
RVFL [43]	89.67	95.67	83.67	85.42	95.67	90.25	89.47
dRVFL [29]	91.17	91.67	90.67	90.76	91.67	91.21	91.17
Proposed Algorithm	96.50	95.33	97.67	97.61	95.33	96.46	96.49

**Table 3 brainsci-13-00267-t003:** Voxel-wise variations in WM, GM, and CSF volumes with *p*< 0.05.

Region of Interest	Anatomical Region	Voxels	T-Value	Z-Value
WM	Left cerebrum Internal Ventricle	1363	6.90	6.21
	Right cerebrum Insula	340	4.83	4.56
	Right cerebrum Temporal lobe	41	3.70	3.57
GM	Left cerebrum Extra -Nuclear	12	4.82	4.56
	Right cerebrum Temporal lobe	27	4.64	4.40
	Left cerebrum claustrum	6	3.47	3.36
CSF	No Clusters are identified			

## Data Availability

The raw data for the proposed work have been accessed from publicly available Open Neuro Dataset.

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
