# Peer review of "Investigating White Matter Abnormalities Associated with Schizophrenia Using Deep Learning Model and Voxel-Based Morphometry"

_brainsci, 2023, doi:10.3390/brainsci13020267_

Round 1

Reviewer 1 Report

The present study related to the identification of abnormalities associated with schizophrenia using a deep learning model and voxel-based morphometry is interesting and overall well-designed. However, edits must be made before being considered for publication in Brain Sciences.

Edits, comments and suggestions :

1.     Authors could combine the introduction (1) and the literature review (2).

2.     The authors could explain more the originality of their study compared to the others.

3.     Lines 17-21 : Authors should add references on schizophrenia.

4.     Lines 26-28 : Include more recent references.

5.     Lines 39-40 “MRI has found widespread application in recent years for SCZ diagnosis”. “The application of ML classifiers for SCZ diagnosis” References are needed.

6.     Line 42 : “CT scans” Need to write full name.

7.    Lines 67-68 : “neuropathological abnormalities have been found”, “neural substrates of connectivity play a key role in disease development” References are needed.

8.     Line 107 : “GAN” Need to write full name.

9.     Line 122 : “in [25]”, Put the name of the author or “in a recent study”.

10.  Line 127: “eighty-six SCZ patients” Put in number.

11.  Line 130 : “in [27]”, Put the name of the author or “in a recent study”.

12.  For the figures : A short description of what is contained in each figure is missing.

13.  Figure 1 : “NIfTI” need to write full name.

14.  Lines 148-149 : The authors must indicate the age, gender of subjects and date of publication in openneuro.

15.  Line 152 : the figure number is not written.

16.  It could be better to insert figure 2 in Part 3.2.

17.  It could be better to insert figures 3 and 4 in Part 3.3.

18.  What is the value of MAE (mean absolute error) in this study compared to other methods in the literature ?

19.  Images in Figure 6 could be improved to be more visible.

20.  a), b), c) are missing in the Figure 7.

21.  a), b), c) are missing in the Figure 8.

22.  It could be better to insert figure 8 in part 4.3.

23.  It could be better to insert Table 3 after line 259 before “Fig 9 shows ….”

24.  Images in Figure 9 could be improved to be more visible.

25.  Discussion part should be in a separate part and must be developed and enriched by references. Authors should discuss their results compared to previous studies.

Reviewer 2 Report

The manuscript entitled “Investigating White Matter Abnormalities Associated with Schizophrenia using Deep Learning Model and Voxel-Based Morphometry” has been investigated in detail. The topic addressed in the manuscript is potentially interesting and the manuscript contains some practical meanings, however, there are some issues which should be addressed by the authors:

1)      In the first place, I would encourage the authors to extend the abstract more with the key results. As it is, the abstract is a little thin and does not quite convey the interesting results that follow in the main paper. The "Abstract" section can be made much more impressive by highlighting your contributions. The contribution of the study should be explained simply and clearly.

2)      The “Introduction” section needs a major revision in terms of providing more accurate and informative literature review and the pros and cons of the available approaches and how the proposed method is different comparatively. Also, the motivation and contribution should be stated more clearly.

3)      What makes the proposed method suitable for this unique task? What new development to the proposed method have the authors added (compared to the existing approaches)? These points should be clarified.

4)      “Results and Discussion” section should be edited in a more highlighting, argumentative way. The author should analysis the reason why the tested results is achieved.

5)      The authors should clearly emphasize the contribution of the study. Please note that the up-to-date of references will contribute to the up-to-date of your manuscript. The study named "Artificial intelligence-based robust hybrid algorithm design and implementation for real-time detection of plant diseases in agricultural environments; Recognition of COVID-19 disease from X-ray images by hybrid model consisting of 2D curvelet transform, chaotic salp swarm algorithm and deep learning technique" - can be used to explain the method in the study or to indicate the contribution in the “Introduction” section.

6)      How to set the parameters of proposed method for better performance?

7)      The complexity of the proposed model and the model parameter uncertainty are not enough mentioned.

8)      It will be helpful to the readers if some discussions about insight of the main results are added as Remarks.

This study may be proposed for publication if it is addressed in the specified problems.

Round 2

Reviewer 2 Report

All my comments have been thoroughly addressed.